# Electrochemical Performance of Layer-Structured Ni_0.8_Co_0.1_Mn_0.1_O_2_ Cathode Active Materials Synthesized by Carbonate Co-Precipitation

**DOI:** 10.3390/nano12203610

**Published:** 2022-10-14

**Authors:** Byung Hyun Park, Taeseong Kim, Hyerim Park, Youngku Sohn, Jongmin Shin, Misook Kang

**Affiliations:** 1Department of Chemistry, College of Natural Sciences, Yeungnam University, Gyeongsan 38541, Korea; 2Department of Chemistry, Chungnam National University, Daejeon 34134, Korea

**Keywords:** layered Ni-rich NCM, cathode active material, carbonate co-precipitation

## Abstract

The layered Ni-rich NiCoMn (NCM)-based cathode active material Li[Ni_x_Co_(1−x)/2_Mn_(1−x)/2_]O_2_ (x ≥ 0.6) has the advantages of high energy density and price competitiveness over an LiCoO_2_-based material. Additionally, NCM is beneficial in terms of its increasing reversible discharge capacity with the increase in Ni content; however, stable electrochemical performance has not been readily achieved because of the cation mixing that occurs during its synthesis. In this study, various layer-structured Li_1.0_[Ni_0.8_Co_0.1_Mn_0.1_]O_2_ materials were synthesized, and their electrochemical performances were investigated. A NiCoMnCO_3_ precursor, prepared using carbonate co-precipitation with Li_2_CO_3_ as the lithium source and having a sintering temperature of 850 °C, sintering time of 25 h, and metal to Li molar ratio of 1.00–1.05 were found to be the optimal parameters/conditions for the preparation of Li_1.0_[Ni_0.8_Co_0.1_Mn_0.1_]O_2_. The material exhibited a discharge capacity of 160 mAhg^−1^ and capacity recovery rate of 95.56% (from a 5.0–0.1 C-rate).

## 1. Introduction

With the rapid acceleration of global warming due to the constant consumption of fossil fuels, unexpected environmental disasters have been occurring worldwide. Effecting adequate international-level responses to climate change-related issues has become a noteworthy objective. As a first step in this context, the global community has established a consensus with respect to the use of renewable energy as an alternative to fossil fuels. Lithium-ion batteries (LIBs) are the most promising candidates for storing renewable energy sources due to their high storage capacity [1,2]; thus, considerable attention has recently been devoted to the development of new materials and performance improvement of LIBs. Among the four core components of LIBs—the anode, cathode, separator, and electrolyte—the cathode material acts as the strongest limiting factor for the LIB performance [3]. Cathode materials, which comprise metal oxides or phosphates as components, have diverse structures such as layered (LiCoO_2_ [4], Li(Ni, Co, Mn)O_2_ [5], and Li(Ni, Co, Al)O_2_ [6]), spinel (LiMn_2_O_4_ [7]), and olivine configurations (LiFePO_4_ [8]) that likely contain Li ions in the lattices. Improving the cathode materials is the objective for achieving high energy densities, long lifespans, excellent thermal stability, and low prices [9]. In particular, NiCoMn (NCM) is a notable cathode material comprising Ni (high concentrations), Co, and Mn that exhibits a higher energy density than that of other cathode materials, such as LiCoO_2_, LiFePO_4_, and LiMn_2_O_4_ [10]. In the NCM cathode material, Ni promotes a high level of delithiation through two-electron transfer via the redox reaction involving Ni^2+^/Ni^3+^ and Ni^3+^/Ni^4+^, thereby enabling a cell discharge capacity of greater than 200 mA h g^−1^ [11]. Mn improves the structural and thermal stability of Ni by hindering irreversible side reactions with the electrolyte in the electrochemically inactive Mn^4+^ state; this state is where the 3d-e_g_ orbital is emptied [12]. Co improves the electrochemical properties through the redox reaction of Co^3+^/Co^4+^ and maintains a stable cycle life [13]. Thus, more electric energy can be generated when a battery of the same size is used [14]. However, although Ni is suitable for the momentary release of a small amount of energy, it is a very unstable material. Hence, a higher Ni content has been reported to cause most battery fires, resulting in a shorter lithium-ion NCM (LiNCM) battery life [15,16]. There are a few disadvantages of NCMs. First, Ni ions exist at the Li sites, which may result in a stoichiometric shortage of Li^+^. Second, during charging, Ni^2+^ attached to the Li plane is oxidized to Ni^3+^, which transforms the NCM crystal structure. Eventually, the insertion of Li^+^ is suppressed, resulting in polarization loss during the initial discharge [17]. Third, Jahn–Teller distortion easily occurs in oxidized low-spin (t_2g_^6^e_g_^1^) Ni^3+^, which increases electrode resistance and decreases capacity [18]. Therefore, to improve the capacity reduction, the Jahn–Teller distortion should be minimized. Finally, NCMs have low thermal stability; in particular, as the cycle progresses, the NCM layer structure is transformed into a spinel structure, making the diffusion of Li^+^ challenging [19]. To compensate for the limitations of LiNCM, layered LiNi_x_MO_2_ (LNM, 0.8 ≤ x ≤ 0.95) has been marked as an alternative [20]. In particular, aluminum (Al)-doped LiCMA has a high price competitiveness and discharge capacity and has recently been in the spotlight as a cathode material for mid- to large-sized secondary batteries [21]. However, LiCMA cathode materials have disadvantages, such as thermal instability due to the collapse of the layered structure during overcharging, low lifespan and rate characteristics, and an increase in the internal impedance as the cycle progresses [22]. Many ongoing studies have aimed to overcome these limitations, and metal doping or surface coating is the most promising method [23,24]. For example, doping boron (B) with small ions improves the electrochemical stability of the LiCMA structure by changing the lattice gap and deforming the electronic structure [25]. In addition, LiNCMA doped with Co at a size similar to Ni can partially replace Ni in the octahedral lattice by bonding with O. Consequently, this change can improve the thermal and structural stabilities and reduce the resistance of electrons or ions [26]. However, although LiNCM exhibits high-capacity electrical activity, the thermal instability and short lifespan of LiNCM compared with conventional cathode active materials are still observed, despite several efforts to alleviate the issue.

Moreover, solid-phase and co-precipitation methods have been used as general manufacturing methods to produce positive electrode materials. However, the solid-phase method has the disadvantages of high temperature, long manufacturing time, and high impurity inflow during mixing, making it challenging to obtain a uniform composition [27]. The co-precipitation method involves precipitating a precursor by preparing an aqueous solution containing Ni, Co, Mn, and sodium hydroxide to adjust the pH and a chelating agent as a complexing agent. The obtained precursor is mixed with Li salt, and a cathode active material is obtained by the calcination [28]. In other words, hydroxide co-precipitation helps overcome the disadvantages of the solid-phase method by obtaining a uniform material composition and characteristics. However, because the particle size of the precursor affects the size of the active material, the number of process variables during the synthesis increases. Consequently, process optimization becomes complicated, time-consuming, and labor-intensive [29]. This method also readily oxidizes Mn^2+^ to Mn^3+^ to form manganese oxidehydroxide (MnOOH) and causes a deviation from the desired stoichiometry. Thus, the hydroxide precursor has an irregular shape and low density, resulting in a low energy density and inadequate electrochemical performance [30]. Carbonate co-precipitation has emerged as an alternative method for producing transition metal (Ni, Co, Mn) precursors [31]. This method maintains the oxidation state of all transition metals at 2+ and can be performed in an environment with a harsher pH than that of the general hydroxide process.

This study aimed to control the layered structure of the cathode material through its synthesis method, that is, a hydrothermal method, using LiNi_0.8_Co_0.1_Mn_0.1_O_2_ (NCM811) crystals obtained via sintering in air at a high temperature. The NCM811 samples were prepared using various synthesis factors that affect the physical properties of the cathode material, such as the precursor type (with or without the NiCoMnCO_3_ precursor prepared by carbonate co-precipitation), concentration of urea, type of Li source, sintering temperature, sintering time, and Li content. The charging and discharging of the NCM811 samples were tested at a rate of 0.1 C to determine the conditions with the best efficiency and establish the optimal synthesis conditions for the NCM811-layered structure. Finally, we tested the NCM811 synthesized under optimal synthetic conditions for a long duration at 0.1 C to test its durability and lifespan. At this stage of investigation, we identified the factors that had the most significant effect on the lifespan of the cathode material, and we offered supplementary measures.

## 2. Materials and Methods

### 2.1. Preparation and Characterization of Cathode Active Materials

The molar ratio of Ni:Co:Mn in NCM was 0.8:0.1:0.1. NiCoMnCO_3_, a precursor of NCM, was first prepared by hydrothermal synthesis (Figure 1). Ni(CH_3_COO)_2_·4H_2_O, Co(CH_3_COO)_2_·4H_2_O, and Mn(CH_3_COO)_2_·4H_2_O, which are acetate salts (Junsei Co., Tokyo, Japan), were used as sources of the metals, and the amounts appropriate to the molar ratio of each metal were added into 60.0 mL of distilled water. Urea (CO(NH_2_)_2_; Junsei Co., Tokyo Japan) was added to the solution at different concentrations (1.0, 1.5, 2.0, 2.5, and 3.0 M) and stirred for 3 h. This solution was placed in 100 mL of Teflon-lined stainless steel and was maintained at 200 °C for 10 h. The obtained solid material was washed with distilled water and ethanol until the pH reached 7, where it was subsequently dried in a dryer set at 80 °C for 24 h to obtain a NiCoMnCO_3_ precursor. The NiCoMnCO_3_ precursor and Li source of either Li_2_CO_3_ or LiOH·H_2_O were added at molar ratios of 1:1.1, 1:2, 1:3, 1:4, 1:5, 1:6, and 1:7 and were kept for 12 h. The mixture was uniformly mixed by a ball-milling method. The mixture was pelleted at 400 psi and pre-calcined in air at 480 °C for 6 h in a tube furnace. Subsequently, the sintering temperature was increased to 700, 750, 800, 850, and 900 °C and maintained in an O_2_ atmosphere for 25 h. Consequently, black LiNi_0.8_Co_0.1_Mn_0.1_O_2_ (NCM811) was obtained.

The synthesized NiCoMnCO_3_ precursor and NCM811 crystals were identified using a powder X-ray diffraction analysis (XRD, model MPD, PANalytical, Malvern, UK). Using the mentioned diffractometer with a Cu Kα radiation of 40.0 kV and 30.0 mA, the 2θ angles were measured in the range of 20–90° at a scan rate of 5° min^−1^. The shapes of the NiCoMnCO_3_ precursor and NCM811 were investigated using a scanning electron microscopy analysis (SEM, S-4100, Hitachi, Tokyo, Japan). Quantitative and qualitative analyses of the synthesized NiCoMnCO_3_ precursor and elements constituting NCM811 were performed using inductively coupled plasma spectroscopy (ICP-AES, OPTIMA 8300, Perkin Elmer, Waltham, MA, USA).

### 2.2. Electrochemical Performance Evaluation

To evaluate the electrochemical characteristics, the half-cell performance was confirmed using a 2032 coin-type cell assembled in an Ar (99.999%) atmosphere in a glove box, wherein moisture and oxygen were controlled to 1 ppm or less. Assembly of the coin cell is described in Figure 2. Regarding the preparation of the electrode, we assigned mass ratios of 8.0:1.0:1.0 to the cathode electrode active material (NCM811), conductive material (carbon black), and binder (polyvinylidene), respectively. This mixture was placed in a N-Methyl-2-Pyrrolidone (NMP) solvent and dispersed to form a slurry. After the slurry was uniformly dispersed, it was applied to aluminum foil as a current collector. Here, the electrode loading and electrode density were controlled at 12.0 mg/cm^2^ and 3.0 g/cm^3^, respectively. Before cell assembly, the electrode was vacuum dried at 120 °C for 12 h. Metal Li was used as the cathode electrode, polypropylene was used as the separator, and 1.0 M of LiPF_6_ (in 1.0 M of ethylene carbonate:1.0 M of dimethyl carbonate (by volume) (PanaX etec. Co., Busan, Korea) was used as the electrolyte.

Battery test equipment (WBCS3000, WonATech Co., Seoul, Korea) was used for the constant current charge/discharge experiments of NCM811. The charging/discharging of the coin cell was performed in a voltage range of 2.7–4.3 V at a 0.1 C-rate. The electrode lifetime was tested for 100 cycles in the range of 2.7–4.3 V at a 0.1 C-rate, and the rate characteristics were evaluated at a 0.1–5.0 C-rate.

## 3. Results and Discussion

### 3.1. Characteristics of NiCoMnCO_3_ Precursors Synthesized Using Different Urea Concentrations

When the added urea is completely decomposed, carbonate precursors such as NiCoMnCO_3_ are formed and precipitated, as expressed in Equation (1), in accordance with the literature [32]:CO(NH_2_)_2_ + 2H_2_O ↔ 2NH_4_^+^ + CO_3_^2−^(1)

The minimum molar ratio we used between the transition metals (Ni, Co, and Mn) and urea was 1.0:1.0. However, as shown in Equation (1), urea decomposes into two ammonium (NH_4_^+^) ions and one carbonate (CO_3_^2−^) ion in an aqueous solution. These compounds combine with divalent transition metals to undergo carbonate co-precipitation and ammonia complex formation, which are competing reactions [33]. Thus, to advantageously prepare a 1.0 M divalent transition metal carbonate, the concentration of urea should be at least two times greater than the 1.0 M concentration of the added metal. Accordingly, in this study, the urea concentration was varied from 1.0 to 3.0 M when synthesizing the NiCoMnCO_3_ precursor.

Figure 1a presents the XRD pattern of the NiCoMnCO_3_ precursor synthesized by changing the concentration of urea to 1.0, 1.5, 2.0, 2.5, and 3.0 M at 200 °C for 24 h. Peaks corresponding to the (012), (104), (110), (113), (202), (024), (116), (018), (122), and (214) diffraction planes of the rhombohedral crystal structure of the NiCoMnCO_3_ precursor were observed at 2θ = 25°, 32°, 38°, 43°, 47°, 51°, 54°, 62°, 67°, and 72°, respectively (JCPDS Card No. 00-012-0771, 01-078-0209, 01-085-1109). The NiCoMnCO_3_ precursor synthesized by adding 1.0 M of urea exhibited a broad peak in the (104) diffraction plane. However, in the NiCoMnCO_3_ precursor synthesized by adding 2.0 M or more of urea, the crystallinity increased, and crystal planes were clearly observed. The excellent crystallinity of the precursor indicates that the NCM811 cathode material synthesized using this precursor may also be crystallographically stable and exhibit excellent electrical performance. SEM images of the NiCoMnCO_3_ precursors (Figure 1b) indicated that they exhibited regular elliptical shapes, with sizes ranging from 2.0 to 3.0 μm. In particular, the NiCoMnCO_3_ precursor synthesized using 1.0 M of urea had elliptically shaped agglomerates of amorphous particles. However, the particles in the carbonate precursors synthesized using ≥1.5 M of urea were present in the form of rectangular plate-like layers. In particular, the plate-like layers of the carbonate precursor synthesized using 1.5 M of urea were considerably thick. However, the carbonate precursors synthesized using higher urea concentrations (2.0 and 2.5 M) had similar but extremely thin plate-shaped particles. Significantly thick plate-shaped particles were observed in the carbonate precursor synthesized sample using 3.0 M of urea; these plates were similar to those of the precursor synthesized using 1.5 M of urea. Overall, these results indicate that the thickness of particles in the carbonate precursor varied according to the urea concentration, which probably affected the insertion of Li.

Table 1 lists the results obtained from an ICP elemental analysis performed to confirm the molar ratio of the transition metals present in the NiCoMnCO_3_ precursor. As previously mentioned, the CO_3_^2−^ ions generated via the complete decomposition of urea must be combined with a transition metal ion to form a precipitate of the NiCoMnCO_3_ precursor. In this case, the solubility product constant (K_sp_) related to the precipitation was 1.3 × 10^−7^ for nickel carbonate (NiCO_3_), 1.0 × 10^−10^ for cobalt carbonate (CoCO_3_), and 5.0 × 10^−10^ for manganese carbonate (MnCO_3_). CoCO_3_ precipitation is the most advantageous process because its K_sp_ is the lowest, whereas NiCO_3_ precipitation is the least favorable process. Therefore, a sufficiently high concentration of CO_3_^2−^ ions is required to ensure the complete precipitation of NiCO_3_. In other words, 2.0 M or more of urea was required for the sufficient NiCO_3_ precipitation of 1.0 M. When using 1.0 M of urea, only approximately 86.4% of the input amount was precipitated with NiCO_3_. Alternatively, when using 1.5 M of urea, 3% of the input amount did not precipitate. Eventually, adding 2.0 M of urea resulted in the precipitation of almost 100% NiCO_3_. However, the NiCO_3_ precipitation decreased again when 2.5 M or more of urea was added; this is because the NH_4_^+^ decomposed in urea reacts with Ni^2+^ to form [M(NH_3_)_n_]^2+^, which hinders the precipitation of NiCO_3_ to a certain extent [34]. Therefore, we concluded that the NiCoMnCO_3_ precursor was synthesized in an optimal state when prepared using a molar ratio of 1:2 for the transition metal (M) to urea. The NCM811 cathode active material electrode prepared using the NiCoMnCO_3_ precursor obtained from 2.0 M of urea exhibited charge and discharge efficiencies of 179.48 and 159.81 mA h g^−1^, respectively.

### 3.2. Characteristics of NCM811 According to Type of Li Source

The urea concentration (2.0 M) and molar ratio of Li and metal (M) were kept constant when investigating the optimal Li source (LiOH or Li_2_CO_3_) and calcination time for the synthesizing of the NCM811 cathode using a NiCoMnCO_3_ precursor. Figure 2a shows the XRD pattern of the NCM811 synthesized using LiOH as the source of lithium and sintering for 15 h at each temperature. Except for the case of NCM811 sintered at 700 °C, all of the NCM811 particles featured two separate peaks corresponding to the (108) and (110) planes, confirming that they had a layered α-NaFeO_2_ (R-3m) structure [35]. In particular, a division of the double lines of (006)/(102) and (108)/(110) was observed in the NCM811 particles sintered at higher temperatures, indicating a more stable and highly crystalline layered structure [36]. Meanwhile, the intensity ratio of the (003) and (104) planes, which are two prominent peaks, that is, I_(003)_/I_(104)_, is regarded as an essential indicator of cation mixing in the crystal lattice [37]. In instances of small I_(003)_/I_(104)_ ratios, other ions occupy the Li region during cation mixing, affecting the performance of Li batteries [38], with cation mixing generally occurring when I_(003)_/I_(104)_ is less than 1.2. The value of this intensity ratio increased as the calcination temperature increased from 700 to 900 °C, which indicated the well-ordered layered structure of all NCM811 particles in the absence of cation mixing, except for the NCM811 synthesized at 700 °C, which exhibited intensity ratios of ≤1.2. Figure 2b shows the results of the charging and discharging tests using NCM811 synthesized at different calcination temperatures with a Li source of LiOH. The tests were performed at a rate of 0.1 C in the potential range of 2.7–4.3 V. The discharging efficiencies of 700LiOH-NCM811, 750LiOH-NCM811, 800LiOH-NCM811, 850LiOH-NCM811, and 900LiOH-NCM811 were 142.19, 135.54, 153.52, 154.71, and 145.97 mA h g^−1^, respectively. 850LiOH-NCM811 calcined at 850 °C exhibited the largest initial discharge capacity. Based on the XRD analysis, the 850LiOH-NCM811 particles exhibited the least cation mixing and had a well-ordered structure. In contrast, the discharge capacity decreased for the NCM811 particles calcined at 900 °C, and the discharge efficiency of the NCM811 particles sintered at 700 °C was higher than that of the NCM811 particles calcined at 750 °C. This decrease in discharge efficiency at a specific temperature is because of a decrease in performance caused by cation mixing, whereby Ni^2+^ ions enter the Li^+^ sites. Therefore, the optimal calcination temperature for the NCM811 particles was 850 °C. Figure 2c shows the XRD patterns of the NCM811 particles prepared by sintering the NiCoMnCO_3_ precursor obtained using Li_2_CO_3_ as the Li source for 15 h at various calcination temperatures. Similar to the NCM811 particles obtained using LiOH as the Li source, all of the NCM811 samples, except for NCM811 sintered at 700 °C, featured a layered α-NaFeO_2_(R-3m) structure. In addition, the division of the double line peaks of (006)/(102) and (108)/(110) became prominent as the calcination temperature increased. This result indicates that NCM811 has a more stable and highly crystalline layered structure. In addition, the values of I_(003)_/I_(104)_, an essential indicator of cation mixing, were 1.19, 1.34, 1.60, 1.81, and 1.91 for the NCM811 particles calcined at 700, 750, 800, 850, and 900 °C, respectively. As the calcination temperature increased, the intensity ratio also increased. Cationic mixing is expected in the 700Li_2_CO_3_-NCM calcined at 700 °C, which exhibited an intensity ratio of 1.2 or lower. However, for all of the remaining NCM811 particles, this intensity ratio was 1.2 or higher; therefore, it can be predicted to have a well-ordered structure without the cation mixing of Li_2_CO_3_. Figure 2d shows the charging and discharging test results for the NCM811 particles obtained using Li_2_CO_3_ as the lithium source. The discharge efficiencies of 700Li_2_CO_3_-NCM, 750Li_2_CO_3_-NCM, 800Li_2_CO_3_-NCM, 850Li_2_CO_3_-NCM, and 900Li_2_CO_3_-NCM were 159.34, 150.06, 157.88, 159.81, and 156.06 mA h g^−1^, respectively. The 900Li_2_CO_3_-NCM particles calcined at 900 °C showed a lower discharge efficiency than that of the 850Li_2_CO_3_-NCM particles calcined at 850 °C, which exhibited the highest value among the specimens, similar to the observations made when LiOH was used as the Li source. The highest initial discharge capacity of the 850Li_2_CO_3_-NCM particles calcined at 850 °C is likely due to having cation mixing to the lowest extent among the specimens and due to having a well-ordered structure, as shown by the XRD results. However, the difference between the charging and discharging capacities based on the calcination temperature did not increase when Li_2_CO_3_ was used as the Li source rather than LiOH. This indicates that Li moves smoothly between the positive and negative electrodes during charging and discharging; therefore, Li_2_CO_3_ is the optimal Li source compared with LiOH.

Figure 3 shows the SEM image of the NCM811 cathode material presented in Figure 2c. The NCM811 particles formed by the intercalation of Li ions between the angular layers of the NiCoMnCO_3_ precursor exhibited a pinecone shape, wherein tiny particles were densely aggregated in an elliptical shape. Each pinecone-shaped particle has a size of 10 × 20 μm (width × length) and is uniformly grown and distributed at a calcination temperature of 750–850 °C. However, fragments from the broken oval pinecone particles and larger amorphous particles were nonuniformly mixed in 900Li_2_CO_3_-NCM calcined at a high temperature of 900 °C. The nonuniformity of these particles creates voids in the cathode electrode, which hinders the diffusion of Li ions; this, in turn, lowers the Li transfer rate, which can cause performance degradation during repeated charge/discharge cycles [39]. Thus, the decrease in the discharge capacity of the 900Li_2_CO_3_-NCM particles may be attributed to the nonuniformity of the particles caused by voids, as evidenced by the SEM image. In addition, the SEM images of 700Li_2_CO_3_-NCM calcined at 700 °C show that the particles were also partially nonuniform compared with those calcined at different temperatures. This outcome indicates a high initial discharge efficiency; however, the partial nonuniformity of these particles results in cation mixing, as described based on the XRD results. Furthermore, it is predicted that Ni ions enter Li vacancies based on the repetition of charge/discharge cycles and cause structural collapse, resulting in a performance degradation. Thus, the optimal Li source and calcination temperature were determined to be Li_2_CO_3_ and 850 °C, respectively.

### 3.3. Characteristics of NCM811 as a Function of Calcination Time

Figure 4a presents the XRD pattern of the NCM811 prepared by varying the calcination time at 850 °C (the optimal calcination temperature) with the NiCoMnCO_3_ precursors prepared using Li_2_CO_3_ as the optimal Li source. All NCM811 particles showed a layered α-NaFeO_2_(R-3m) crystal structure. The division of the doublets of (006)/(102) and (108)/(110) was observed in the NCM particles calcined for 5 h. However, the peak splitting in the 5 h-NCM was weak, indicating a relatively low crystal stability. The peak splitting of (006)/(102) and (108)/(110) was also observed in the NCM that was calcined for more than 10 h, suggesting that it features a layered structure with excellent crystal stability. Moreover, as the overall peak intensity remained similar from the 10 to 25 h of calcination, it was assumed that the crystal growth did not significantly change after calcination for 10 h or higher. The intensity ratios (I_(003)_/I_(104)_) for the NCM particles calcined for 5, 10, 15, 20, and 25 h to confirm the cation mixing process were 1.91, 1.85, 1.80, 1.82, and 1.98, respectively. All of the NCM811 particles exhibited I_(003)_/I_(104)_ values of 1.2 or higher; this implies that all of the NCM811 particles had well-ordered structures without cation mixing. Figure 4b shows the results of the charge/discharge tests for the NCM811 particles synthesized according to firing time. The charge/discharge experiment was performed at a rate of 0.1 C in the potential range of 2.7–4.3 V. As predicted from the XRD results, the 5 h-NCM particles, which were calcined for 5 h, exhibited a relatively low discharge capacity owing to their low crystal stability. However, for the 10 h-NCM particles, the discharge performance improved owing to the stable layered structure. This discharge capacity remained constant in the 20 h-NCM particles. However, in general, the distribution of positive ions allows for the diffusion and redistribution of Ni and Mn ions via thermal treatment for a longer period of time than when calcined for a short time [40]. In addition, if Ni cations are present on the surface of the crystal without diffusion into the interior of the crystal under short-term thermal treatment, these Ni ions may be converted into nonreactive NiO as the charge/discharge cycles proceed [41]. Therefore, the crystal must have a well-ordered structure without cation mixing to uniformly distribute the Ni in the inside (core) rather than at the surface. Considering that a long firing time is advantageous, 25 h was chosen as the optimal calcination time in this study.

### 3.4. Characteristics of Synthesized NCM811 According to Li/M Molar Ratio

Li has a high vapor pressure and readily evaporates during high-temperature sintering; accordingly, it is necessary to address these characteristic limitations [42]. Therefore, the molar ratio of the transition metal to Li must not be 1:1, but rather be a higher concentration for the Li. To compensate for the small amount of lithium, the molar ratio of the transition metal M:Li was changed from 1:1.01 to 1:1.07, while increasing the molar concentration of Li by 0.01 M. The NCM811 particles were prepared, and the optimal Li content was determined. Figure 5a shows the XRD pattern of the synthesized NCM811 particles with respect to the Li molar ratio. Layered α-NaFeO_2_(R-3m) crystals appeared in all samples, and double line divisions of (006)/(102) and (108)/(110) were observed. The I_(003)_/I_(104)_ values calculated to confirm the cation mixing process were 1.71, 1.88, 1.80, 1.78, 1.81, 1.75, and 1.89 for 1:1.01NCM, 1:1.02NCM, 1:1.03NCM, 1:1.04NCM, 1:1.05NCM, 1:1.06NCM, and 1:1.07NCM, respectively. This implies that all NCM811 particles have a well-ordered structure without a structural collapse caused by cation mixing. Figure 5b shows the charge/discharge test results for the NCM811 particles synthesized at different M:Li molar ratios. The potential range and rate for the charging and discharging test conditions were 2.7–4.3 V and 0.1 C, respectively (similar to the conditions in all previous experiments). At 1:1.01NCM, 1:1.02NCM, 1:1.03NCM, 1:1.04NCM, 1:1.05NCM, 1:1.06NCM, and 1:1.07NCM, the discharge capacities were 146,61, 154.75, 150.72, 146.45, 160.39, 144.06, and 151.16 mA h∙g^−1^, respectively. If the amount of Li added is excessively low, the amount of Li transferred between the electrodes owing to the evaporation of Li remains insufficient, resulting in a low electrical performance. Conversely, if the amount of Li is excessively high, residual Li ions remain, and these Li ions interfere with each other. Eventually, the diffusion of Li^+^ forms a path, resulting in a low discharge capacity during the charging and discharging processes [43]. In this study, the condition for minimizing evaporation and residual Li was determined to be a transition metal M:Li ratio of 1:1.05, which afforded the highest discharge capacity; specifically, a discharge capacity of 160.39 mA h g^−1^ was achieved.

The quantitative analysis results for the elements present in the optimal LiNCM811 sample, as measured using the ICP analysis, are presented in Table 2. The Li:Ni:Co:Mn ratio was 1.022:0.797:0.101:0.102, which was similar to that of the precursor. The sample was stirred and washed in ethylene carbonate/dimethyl carbonate electrolyte solvent for 24 h to determine if Li was exposed on the surface. Thereafter, the quantitative analysis was repeated (Table 3). The results indicated that the Li:Ni:Co:Mn ratio was 1.040:0.790:0.106:0.103, almost identical to that before washing. Thus, it was confirmed that no Li existed outside the NCM811 crystals.

### 3.5. Evaluation of Cycle Performance and Rate Characteristics in NCM811 under Optimal Conditions

In this study, it was found that the optimal synthesis conditions for NCM811 were a molar ratio of 1:2 for the transition metal, M:urea, the usage of Li_2_CO_3_ as the lithium source, a calcination temperature of 850 °C, a firing time of 25 h, and a molar ratio of 1:1.05 for the transition metal, M:Li. Figure 6a presents the isothermal curve for the N_2_ gas adsorption/desorption of the NCM811 particles synthesized under optimal conditions. As the adsorption and desorption curves of nitrogen gas in the NCM811 particles were almost identical, the hysteresis was negligible. This is typical for IUPAC Type III isotherms, which occur in the absence of pores or for a mixture of non-pores and macropores [44], implying a lack of regular pores in the NCM811 particles. A small specific surface area of 2.6 m^2^/g and small volume of 0.015 cm^3^/g were also noted. Moreover, a pore of approximately 23.35 nm also appeared; however, based on the pore volume, it was presumed to be a bulk pore formed via the agglomeration of particles. According to these results, the transfer reaction of Li ions in the battery occurs on the surface of the bulk particles; the bulk pores, rather than the regular pores in the crystal, can be inferred to affect the electrical activity. Figure 6b shows the charge/discharge curve for the test performed in the potential range of 2.7–4.3 V with the optimal NCM811 at a 0.1 C-rate. The charging and discharge capacities were 182.13 mA h g^−1^ and 162.50 mA h g^−1^, respectively. Furthermore, the cycle performance was determined in the potential range of 2.7–4.3 V at a rate of 0.5 C to evaluate the durability (Figure 6c). In the first cycle, the discharge capacity was 151.8 mA h g^−1^, which subsequently decreased to 98.78 mA h g^−1^ after 100 cycles; however, it was confirmed that the capacity retention rate remained at 65.1%. Coulombic efficiency is the ratio between the capacity achieved by charging and the capacity realized by the preceding charging step. This parameter is directly related to the battery life and depends on the temperature and C-rate; in particular, it decreases with increasing cycle time due to the occurrence of self-discharge. The optimal NCM cathode agent synthesized in this study exhibited a coulombic efficiency of 99% at a 0.5 C-rate, which was maintained until the 100th cycle. In general, the Coulombic efficiency of lithium ions is 99% at the 0.05 C-rate and drops to approximately 97% at the 0.5 C-rate [45]. However, the Coulombic efficiency of 99% achieved at the 0.5 C-rate in this study was maintained for a long time. The use of the carbonate precursor possibly reduced electrolyte oxidation at the cathode, resulting in an improved SEI, which stabilized the Coulombic efficiency [46]. In the rate test presented in Figure 6d, the rate of recovery was initially 0.1 C; it gradually increased in the order of 0.2 C, 0.5 C, 1.0 C, 2.0 C, and 5.0 C, and then was decreased to 0.1 C. At rates of 0.2 C, 0.5 C, 1.0 C, 2.0 C, and 5.0 C, the capacities were 96.9%, 89.99%, 84.69%, 77.24%, and 63.86%, respectively, as compared with the initial capacity of 162.50 mA h g^−1^ at the rate of 0.1 C. It should be noted that a capacity recovery rate of 95.56% was maintained in the recovery step from the rates of 5.0 C to 0.1 C.

As shown in Table 4, the discharge efficiencies of previously reported batteries containing NCM cathode materials with similar metal component ratios are considerably higher than those of the NCM materials reported in this study [18,47,48,49,50]. However, their durability is relatively low, which significantly reduces their performance retention rate after 100 cycles; moreover, their recovery power is inadequate (<90%). In contrast, the NCM cathode material synthesized using the Ni_0.8_Co_0.1_Mn_0.1_CO_3_ carbonate precursor in this study exhibited a relatively low charge/discharge efficiency, relatively high capacity retention rate, and high capacity recovery (>95%). This indicates that the Ni_0.8_Co_0.1_Mn_0.1_CO_3_ carbonate precursor has a positive effect on the durability of the NCM cathode material.

The size and thickness of the NCM811 particles prepared in this study were mostly larger than 20 µm; therefore, their shape and size could not be accurately examined using a high-resolution transmission electron microscopy (HR-TEM) analysis. However, HR-TEM images were obtained using partial fragments of the ruptured particles (Figure 7). The (003) lattice plane, which was found to affect the cell performance from the viewpoint of cation mixing via the XRD analysis, was clearly observed with a high resolution. The lattice spacing was determined to be 0.473 nm.

In electrochemical impedance spectroscopy (EIS), a phase difference appears between the input and output values based on the applied signal; this is divided into real and imaginary parts, as shown in Equation (2) [51]. The derived impedance can be plotted as a Nyquist diagram, which is expressed as a complex coordinate system in the frequency domain. Here, the real part is the resistance, and the imaginary part is the capacitance and inductance.
Z = Z_real_ − jWZ_imag_
(2)

The half-coin cell setup used in this study can be modeled as a circuit comprising three resistive components: R_s_, which is the resistance of the external electrolyte; R_ct_ (semicircle, related to charge transfer), which represents the oxidation–reduction reaction of Li ions at the surface of the inner electrode particles and the electrode material interface; and Z_w_ (the Warburg impedance), which is the chemical diffusion resistance induced by intercalation into the crystal structure of the grains. Repeatability can be determined as an important result through an EIS analysis of the batteries, in that identical results should be obtained for the same object under identical conditions. In this regard, almost identical data were obtained for the initial and 20th cycles (Figure 8a). The galvanostatic intermittent titration technique (GITT) was performed after measuring the basic charge/discharge capacity. GITT is a useful and important method for determining thermodynamic, kinetic, and component-dependent parameters. This is a remarkably precise method that can permit accurate measurements of both current and time. The GITT data of the stored NCM material during the first charging/discharging cycle (Figure 8b) were obtained as time-dependent voltage and current curves for a single current pulse. When acquiring the GITT data, an open circuit step is followed by a constant-current step, which yields information related to the thermodynamics and diffusion coefficients. The diffusion coefficient can be obtained using Equation (3) [52], where n_m_ is the number of moles; V_m_ is the molar volume of the electrode; S is the electrode/electrolyte contact area; ∆E_s_ is the change in the steady-state voltage; and ∆E_t_ is the change in voltage during the pulse, which can be obtained by removing the IR-drop section.
(3)$$D=\frac{4}{\pi \tau }{\left(\frac{n_{m}V_{m}}{S^{2}}\right)}^{2}{\left(\frac{\Delta E_{8}}{\Delta E_{t}}\right)}^{2}$$

D_Li+_, the diffusion coefficient of lithium ions in the coin cell containing the optimized NCM cathode material, was calculated to be 2.228 × 10^−11^ cm^2^ S^−1^. This value is slightly lower than the diffusion rate of lithium ions in conventional NCM811, probably because bulk-phase crystals were obtained in the present study rather than pure single crystals [53].

Figure 3 shows a schematic of the charge/discharge mechanism that occurs in a full-cell battery. The cell comprised a yC anode (commercialized graphitic carbon) and the LiNCM811 cathode synthesized in this study. During charging, Li^+^ ions escape from the LiNCM811 cathode material and move to the electron-rich yC anode material. A nonspontaneous reaction occurs at this juncture, thereby oxidizing the cathode material to Li_1−x_NCM811 and reducing the anode material to Li_x_C_y_. Conversely, a spontaneous reaction occurs during discharge, with Li^+^ naturally moving from the high-potential anode material to the low-potential cathode material through the electrolyte. Simultaneously, electrons are also collected by the cathode material along the external conductor; thus, the cathode material gains electrons and is reduced to LiNCM811, whereas the anode material is oxidized to yC [54].

## 4. Conclusions

In this study, a NiCoMnCO_3_ carbonate precursor was synthesized under the optimal conditions of a 8:1:1 Ni:Co:Mn ratio by controlling the molar concentration ratio of urea to metal at 1:2. The carbonate precursor was mixed with a Li source to synthesize LiNi_0.8_Co_0.1_Mn_0.1_O_2_(NCM811). The optimal NCM811 was synthesized by changing the type of Li source, calcination temperature, calcination time, and metal to Li molar ratio. The optimal conditions were determined as: a Li_2_CO_3_ lithium source; calcination temperature of 850 °C; calcination time of 25 h; and metal to Li molar ratio of 1:1.05. The NCM811 particles synthesized under these optimal conditions demonstrated the lowest amount of cation mixing and highest crystal stability with respect to the layer. The optimal NCM811 particles exhibited a capacity retention rate of 65.1%, even after 100 charging and discharging cycles at the rate of 0.5 C. Evaluations of the rate characteristics revealed a capacity retention rate of 63.86%, obtained from the rates of 0.1–5 C. In addition, the capacity recovery rate from 5.0 C to 0.1 C was 95.56%. In particular, the possibility of suppressing the conversion to NiO, which could potentially occur on the surface of crystals, was confirmed by comparing the XRD patterns obtained before and after 1000 CV cycles. The results suggest that the improvement in the performance of NCM811 using LiCO_3_ could be further enhanced as the charging and discharging cycles progressed because the formation of dendrites was significantly reduced when LiCO_3_ was used as the Li source.

## Data Availability

Not applicable.

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
