# Peer review of "Electrochemical Performance of Layer-Structured Ni0.8Co0.1Mn0.1O2 Cathode Active Materials Synthesized by Carbonate Co-Precipitation"

_nanomaterials, 2022, doi:10.3390/nano12203610_

Round 1

Reviewer 1 Report

In this manuscript, the authors report the synthesis of various Li1.0[Ni0.8Co0.1Mn0.1]O2 layered structures and investigate their electrochemical performance. The role of urea, lithium source, sintering temperature, sintering time and a metal:Li molar ratio during the synthesis process were studied. Moreover, the optimized Li1.0[Ni0.8Co0.1Mn0.1]O2 showed excellent electrochemical performance.

However, the charge-discharge mechanisms of NCM811 are unclear. Overall, to make the paper better, the manuscript will need to be considerably revised to meet the scientific papers's standards while also catching the readers' attention. The following are some of the concerns that must be fixed. Followings are some of my specific comments:

1. A major refinement is necessary to soften the presentation and to minimize the ambiguity. For example, in page 2, “The co-precipitation method uses an aqueous solution containing Ni, Co, Mn, and sodium hydroxide as a co-precipitant and a chelating and complexing agent for simultaneous precipitation.” “a metal:Li molar ratio of 1:1.05.”

2. In page 2, “The synthesis method is ascertained by the sol-gel method,” First question: Why did the authors use sol-gel method? Second question: in the “Abstract”, “NiCoMnCO3 precursor prepared by carbonate co-precipitation”, however, in the department of 2.1, “a precursor of NCM, was first prepared by hydrothermal synthesis”, which method was used to synthesis the precursor and/or cathode material in this paper?

3. More discussions on the morphology of NiCoMnCO3 precursor should be added, especially for using 1.5 M urea (or higher).

4. In page 6, “……obtained from 2.0 M urea featured a charge efficiency of 179.48 and a discharge efficiency 190 of 159.81 mAhg-1.” The charge-discharge curves should be given to support this result.

5. In page 7, “The discharge efficiencies of 700Li2CO3-NCM, 750 Li2CO3-NCM, 800 Li2CO3-NCM, 850 Li2CO3-NCM, and 900 Li2CO3-NCM are 159.34, 243 140.06, 153.52, 147.88, 159.81, and 156.06 mAhg-1, respectively.” Please correct it.

6. “The NCM811 particles formed by the intercalating Li ions between the angular layers……”, how to prove it?

7. The I(003)/I(104) values of all the prepared materials in this paper lower than 1.2, it was wrong imply that all the NCM811 particles had a well-ordered structure, without cation mixing.

8. In Figure 4b, the charge-discharge efficiencies of NCM811 prepared at 25 h is higher than the others, how about 30 h (or higher sintering time)?

9. The charge-discharge mechanisms of NCM811 should be studied.

10. What is the EIS and GITT?

Reviewer 2 Report

Recommendation: major revision.

Comments: In this work, the authors prepared various Li1.0[Ni0.8Co0.1Mn0.1]O2 layered structures were synthesized via a co-precipitation method. The optimal conditions for preparing Li1.0[Ni0.8Co0.1Mn0.1]O2 were using a NiCoMnCO3 precursor with Li2CO3 as lithium source at a sintering temperature of 850 °C for 25 h. The structure and physicochemical properties of as-prepared materials were studied in detail with suitable techniques and reasonably explained. The electrochemical performance of these Li1.0[Ni0.8Co0.1Mn0.1]O2 cathodes for LIBs is investigated. The experiment data relevant to the Ni-rich layered Li1.0[Ni0.8Co0.1Mn0.1]O2 cathode offered in this manuscript are sufficient to support the conclusion. So, I recommend that this manuscript can be accepted for publication in Nanomaterials after major revision.

1.    The introduction of this paper needs to make a strong argument about the impact and novelty of the work further. So, the introduction should enrich some related cathodes in this section (such as Nanomaterials, 2022, 12, 1888; RSC Adv., 2015, 5, 84673).

2.    The thickness of the loaded active materials can be evaluated?

3.    The authors better offer the TEM, high-resolution TEM image of the prepared Li1.0[Ni0.8Co0.1Mn0.1]O2 cathodes.

4.    To evaluate these Li1.0[Ni0.8Co0.1Mn0.1]O2 cathodes undergo, the SEM images or TEM after cycles are better offered.

5.    The corresponding culombic effciency of Figures 6c and 6d are better added in the manuscript.

6.    The Li+ diffusion coefficient of Li1.0[Ni0.8Co0.1Mn0.1]O2 better calculated by galvanostatic intermittent titration technique.

7.    The authors better compare the electrochemical performance of Li1.0[Ni0.8Co0.1Mn0.1]O2 with reported Ni-rich layered cathode materials. Such as J. Mater. Chem. A, 2021, 9, 2830; Nat. Commun., 2021, 12, 1; Nano Energy, 2021, 83, 105854.

Reviewer 3 Report

I have read the manuscript “Synthesis and electrochemical performances of NCM811 cathode active materials with layered structures by a carbonate co-precipitation method” submitted to Nanomaterials, MDPI. Energy storage technology is one of the most critical technologies to the development of new energy electric vehicles and smart grids. Among the energy storage technologies, lithium-ion technology is an attractive energy storage device exhibiting excellent efficiency of charge-discharge processes along with high energy density among the available other rechargeable systems. They are an indispensable power source for a variety of portable electronic devices and EVs today. Lithium-ion batteries are now dominating due to their excellent electrochemical performance; however, the aging of materials is a significant problem for rechargeable batteries. Among the available layered Li–Ni-Co–Mn–O composites, the authors have reported NCM811 by a carbonate co-precipitation method to obtain a phase pure product. The novelty of this work lies in cation mixing occurring during synthesis and controlling the particle size and phase-pure product. The weird part of the article is no electrochemical data.

The work in its present form is not publishable and needs some revisions before rendering a final decision. 

The following points need to be considered.

·         The opening remark in section 1 (intro) must be on the batteries and their importance. Then, followed by cathodes, etc.

·         Page 1, section 1, line 26 what are “other cathode materials” mention the materials in detail?

·         Line 29, how higher is high? Please mention the number. (0.6??)

·         Jahn-Teller deformation is mainly connected to spinel LMO associated with Mn. Is it also with Ni?

·         In the introduction section (related to NCM and cathodes for Li-ion batteries), relevant reported works on alternative materials illustrating potential candidates expressed in these works (such as Progress in Solid State Chemistry 62 (2021) 100298; Energies 13 (2020) 1477) needs to be included and discussed.

·         Mn(OH)2 is unstable in an aqueous solution with high pH values and forms a secondary phase of MnOx, which can substantially deteriorate the electrochemical characteristics of the final products. Is this usual in sol-gel synthesis?

·         What is the rationale for choosing sol-gel?

·         How about Al2O3 /MgO coatings reported to generate beneficial properties for NMC811?

·         Section 3.1; Urea is usually used as a fuel in combustion synthesis (as discussed in doi.org/10.1016/j.mtener.2018.08.004) but what is the role of urea in sol-gel synthesis? Please discuss.

·         The charge-discharge curves shown in Figure 2 need to be provided with the C-rates.

·         What is the role of particle size in terms of Li-ion transfer in the electrochemical processes?

·         Has the temperature been optimized?

·         The role of individual cations needs to be described.

·         Section 4, line 405 “by comparing the XPS spectra” is it true? Maybe XRD, not XPS.

·         The obtained results need to be benchmarked with other NMC counterparts.

Round 2

Reviewer 1 Report

In this manuscript, the authors report the synthesis of various Li1.0[Ni0.8Co0.1Mn0.1]O2 layered structures and investigate their electrochemical performance. The role of urea, lithium source, sintering temperature, sintering time and a metal:Li molar ratio during the synthesis process were studied. Moreover, the optimized Li1.0[Ni0.8Co0.1Mn0.1]O2 showed excellent electrochemical performance. In my opinion, the article can be published in present form.

Reviewer 2 Report

The quality of the manuscript has been improved significantly.  It is now publishable.

Reviewer 3 Report

I have read the revised part of the manuscript and responses made by the authors to the reviewer's queries. The authors did a fair bit of a job, and it appears the revised version is suitable to publish.